# OpenReview forum: "Discriminative Policy Optimization for Token-Level Reward Models"
_ICML.cc/2025/Conference — ICML 2025 poster_

### Official Review · Reviewer_3WSd · 2025-02-24

**Overall Recommendation:** 3

**Summary:**

This paper introduces a method to construct token level reward from pairwise data. It is further shown that such rewards can be applied to methods like PPO or REINFORCE. Experiments show that both PPO and REINFORCE can be benefited from the constructed token level rewards and outperforms their counterparts with other baseline reward or reward models.

**Claims And Evidence:**

1.The paper claims that incorporating RL algorithms with Q-RM improve the performance of RL algorithm over the baselines

2. A higher accuracy in reward models does not necessarily leads to improved policy performance. A fine-grained credit assignment, even not accuracy enough, can still benefit RL training

**Essential References Not Discussed:**

I do not see any missing of significant previous related work.

**Experimental Designs Or Analyses:**

The experiment covers a wide range of tasks and datasets. Some concerns are listed below

1. It seems that for both PPO and REINFORCE, the author report the detailed training hyper-parameter but lacking the details about how the hyperparameters are selected. Also the training details are missing for methods like DPO / SimPO

2. The author report pass@N on math reasoning tasks. However, pass@N is not a regular metric for math problem solving (because usually the users of LLM are not expected to verify the answer).

**Methods And Evaluation Criteria:**

This paper investigates three tasks, mathematical reasoning, reading comprehensive and instruction following. Widely adopted datasets like GSM8K / MATH, QA-feedback and AlpacaEval are used for evaluation.

**Other Comments Or Suggestions:**

N/A

**Other Strengths And Weaknesses:**

N/A

**Questions For Authors:**

N/A

**Relation To Broader Scientific Literature:**

Previous works show that the the resulting policy of DPO can be used as token-level reward construction. This paper instantiate a construction procedure and used it for RL training and also conducted experiments to verify its benefit.

**Theoretical Claims:**

The proof looks correct though I did not check every details.

---

> ### Author Rebuttal · Authors · 2025-04-01
>
> Thank you for your detailed review and insightful questions on our paper. Below, we address your concerns in detail.
> > Q1: It seems that for both PPO and REINFORCE, the author report the detailed training hyper-parameter but lacking the details about how the hyperparameters are selected. Also the training details are missing for methods like DPO / SimPO
>
> A1：Thank you for your valuable suggestions. We conducted a thorough hyperparameter tuning process to determine the optimal values for our experiments.
>
> - **For REINFORCE-based Methods:** We tuned the learning rate within the range [5e-7, 1e-6, 5e-6] and found that 1e-6 consistently yielded the best performance.
>
> - **For PPO-based Methods:** We explored learning rates in the range [1e-6, 5e-6, 1e-5] and identified 5e-6 as the most effective.
>
> - **For Q-RM:** We searched for the optimal $\beta$ parameter from [0.1, 0.2, 0.5]  and the $\gamma$ parameter from [1.0, 2.0, 5.0], concluding that $\beta = 0.2$ and $\gamma = 2.0$ produced the best results.
>
> - **For Baseline Methods (e.g., DPO and SimPO):** We followed the recommended configurations from their original papers as well as the default settings from widely used repositories (e.g., [Hugging Face’s TRL repository](https://github.com/huggingface/trl)).
>
> We will include these details in the revised version of the paper.
>
>
> > Q2: Pass@N is not a regular metric for math problem solving (because usually the users of LLM are not expected to verify the answer).
>
> A2: Thank you for the thoughtful comment. While we agree that users typically expect a single correct answer, we clarify this concern as follows.
>
> **Pass@N** measures the probability that at least one of N sampled outputs is correct. This metric is particularly useful for assessing a model's best-case capability under stochastic generation. **Pass@N** has become an increasingly popular metric for evaluating LLMs on mathematical tasks (e.g., GSM8K, MATH) [1][2][3], as it reflects the likelihood of generating a correct solution within a limited number of attempts. This provides valuable insights into the model’s reasoning ability and output diversity.
>
> To directly address your concern, our study also reports **Pass@1** alongside **Pass@N**. **Pass@1** corresponds to real-world scenarios where only a single response is returned to the user, making it a practical measure of immediate correctness.
>
> [1] Kimi k1.5: Scaling Reinforcement Learning with LLMs. Arxiv, 2025.
>
> [2] rStar-Math: Small LLMs Can Master Math Reasoning with Self-Evolved Deep Thinking. Arxiv, 2025.
>
> [3] Large language monkeys: Scaling inference compute with repeated sampling. Arxiv, 2024.

---

### Official Review · Reviewer_Hf7f · 2025-03-04

**Overall Recommendation:** 4

**Summary:**

This paper aims to develop a token-level reward mechanism in RLHF. The authors achieve this by decoupling the reward model from language generation and constructing a reward model through the optimization of a discriminative policy. Additionally, they provide a comprehensive theoretical analysis of their method. Experimental results demonstrate the superiority of their approach over existing techniques.

## update after rebuttal
The authors' reply address my concerns, and I decide to keep the score.

**Claims And Evidence:**

Yes

**Essential References Not Discussed:**

None.

**Ethical Review Concerns:**

Please address all weaknesses noted above.

**Experimental Designs Or Analyses:**

The experimental results provide robust evidence that supports the authors’ claims.

**Methods And Evaluation Criteria:**

The proposed method and evaluation criteria make sense.

**Other Comments Or Suggestions:**

None

**Other Strengths And Weaknesses:**

Strengths

1.	The paper is well-written and effectively communicates the key ideas.

2.	The proposed method is clearly presented and easy to follow, making it accessible to readers.

3.	The authors provide a thorough theoretical analysis of their method, adding depth to the study.

4.	The authors conduct extensive experiments, evaluating their approach using various models and baseline methods, which strengthens the empirical validation.

Weaknesses [Minor]

1.	Lines 78–80 mention “under certain assumptions.” These assumptions should be explicitly highlighted and explained in the main body of the paper to ensure clarity.

2.	Including error bars in the experimental results would better demonstrate the robustness of the method and provide more insight into the variability of the outcomes.

3.	The paper lacks human evaluation.

4.	The paper does not provide sufficient details about the hyperparameters required to reproduce the method, which may hinder reproducibility.

**Questions For Authors:**

Please address all weaknesses noted above.

**Relation To Broader Scientific Literature:**

The key contributions of this paper are closely related to token-wise RLHF for LLMs.

**Theoretical Claims:**

Yes, I have checked the theoretical claims. Proposition 3.2. provides bounds for $\mathcal{V}(\tau)$, and the authors provides experimental analysis in Appendix. Proposition 3.3 demonstrates that the optimal Q-function and the logit of optimal policy share a linear relationship.

---

> ### Author Rebuttal · Authors · 2025-04-01
>
> Thank you for your detailed review and insightful questions on our paper. We greatly value your feedback and appreciate your recognition of our work's advancements. Below, we address your concerns in detail.
>
> > Q1: Lines 78–80 mention “under certain assumptions.” These assumptions should be explicitly highlighted and explained in the main body of the paper to ensure clarity.
>
> A1: Thank you for your valuable suggestion.
>
> The assumption in question refers to the trajectory entropy of the optimal policy being close to zero, as stated in Proposition 3.2. This condition constrains $\mathcal{V}(\tau)$ within a narrow range, facilitating the effective training of Q-RM using Eq. 17 in the paper. Furthermore, as demonstrated in Proposition 3.3, computing advantage functions with Q-RM is equivalent to utilizing the optimal Q-functions.
>
> We will ensure that this assumption is explicitly highlighted and more clearly explained in the revised version of the paper.
>
>
> > Q2: Including error bars in the experimental results would better demonstrate the robustness of the method and provide more insight into the variability of the outcomes.
>
> A2: We appreciate the reviewer’s insightful suggestion and agree that statistical indicators can offer a clearer perspective on method stability.
> In our study, we evaluate Q-RM across three task categories: mathematical reasoning (GSM8K, MATH), machine reading comprehension (QA-Feedback), and instruction-following (AlpacaEval 2.0). For GSM8K and MATH, we report **Pass@N** metrics, which inherently account for statistical variation by measuring the probability of obtaining at least one correct sample from N generated completions. These values, computed over multiple sampled responses (e.g., N=8,16), serve as empirical success rates over stochastic rollouts, indirectly capturing performance variance in a manner analogous to confidence intervals.
>
> Following the reviewer’s suggestion, we have now included standard error for QA-Feedback and AlpacaEval 2.0 as follows.
>
> **QA-Feedback:**
> | Method  | Avg | Std Error |
> |-----------|--------|-------|
> | REINFORCE + ORM    | 0.5813 | 0.06 |
> | REINFORCE + DPO-RM    | 0.5581  | 0.06 |
> | REINFORCE + Q-RM    | 0.5917  | 0.07 |
>
> **AlpacaEval 2.0:**
> | Method  | Win Rate (%) | WR Std Error | LC Win Rate (%) |  LC WR Std Error |
> |---------|--------------|----|---|---|
> | REINFORCE + ORM    | 7.72 | 0.86 | 11.00 | 0.47 |
> | REINFORCE + DPO-RM    | 7.96 | 0.88 | 11.87  | 0.50 |
> | REINFORCE + Q-RM    | 9.96 | 0.97 | 14.21  | 0.53 |
>
> More results will be included in the revised paper.
>
>
> > Q3: The paper lacks human evaluation.
>
> A3: Thank you for raising this important point. We would first like to clarify our evaluation strategy.
>
> For instruction-following tasks, we employ AlpacaEval 2.0, which utilizes the LLM-as-a-judge framework. This approach has gained traction as a scalable, cost-effective, and reproducible alternative to traditional human evaluation. Recent studies [1][2][3] have demonstrated that LLM-as-a-judge evaluations strongly correlate with human judgments, particularly when using advanced reference models like GPT-4 as the evaluator. This makes it a practical and reliable substitute for large-scale assessments. Furthermore, AlpacaEval 2.0 incorporates length-controlled win rate (LC win rate) metrics to further reduce bias, enhancing the evaluation’s reliability and interpretability.
>
> We acknowledge the reviewer’s point that human evaluation remains the most direct and interpretable method for assessing LLM performance, particularly for subjective or nuanced outputs. However, due to time constraints during this rebuttal phase, we are unable to complete and present human evaluation results within a few days. We will include these results in the revised version of the paper.
>
> [1] Judging LLM-as-a-Judge with MT-Bench and Chatbot Arena. NIPS 2023.
>
> [2] Length-Controlled AlpacaEval: A Simple Way to Debias Automatic Evaluators. COLM 2024.
>
> [3] Can LLMs Replace Human Evaluators? An Empirical Study of LLM-as-a-Judge in Software Engineering. ISSTA 2025.
>
> > Q4: The paper does not provide sufficient details about the hyperparameters required to reproduce the method.
>
> A4：Thank you for your valuable comment. We provide detailed hyperparameter settings in Appendix D. For baseline methods, we adhere to the recommended configurations from their original papers and the default settings from widely used repositories (e.g., [Hugging Face’s TRL repository](https://github.com/huggingface/trl)). For example, in DPO, the beta parameter is set to 0.1, while in SimPO, beta is set to 1.0 and gamma to 2.0. Moreover, for GRPO, RLOO, and RFT, we generate four responses per instruction, following the approach in [1].
>
> We will further clarify and supplement these details in the revised version of the paper.
>
>
> [1] Back to Basics: Revisiting REINFORCE-Style Optimization for Learning from Human Feedback in LLMs. ACL 2024.

---

### Official Review · Reviewer_sPHu · 2025-03-12

**Overall Recommendation:** 3

**Summary:**

This paper introduces a discriminative Q-function Reward Model (QRM) for token-level credit assignment in RL. Theoretically, the authors show that QRM logits linearly relate to optimal Q-functions, enabling advantage computation without GAE. Empirically, QRM outperforms prior token/sequence-level RMs in tasks like mathematical reasoning and instruction-following when integrated into RL frameworks (PPO, REINFORCE). The work is technically sound, with elegant theoretical derivations and thorough experiments.

**Claims And Evidence:**

**Claim 1**: Equivalence between $Z^*(s_t,a_t)$ and $Q^*(s_t,a_t)$. (Proposition 3.3)

**Feedback**: The connection between Proposition 3.3 and the equivalence claim requires more detailed derivation. For instance:

- Clarify how the linear relationship implies equivalence in practical RL settings.
- Discuss assumptions (e.g., stationarity, policy invariance) needed for this equivalence.

**Claim 2**: Effectiveness across reasoning/QA tasks (Section 5.1).

**Feedback**: While results on GSM8K/MATH are strong, broader validation would strengthen claims:

- For mathematical reasoning task like GSM8K and MATH, authors firstly train Q-RM on their training set and conduct RL training on the same training instructions. The Q-RM is used on in distribution data during RL phase, I suggest RL training on a different OOD instruction dataset, showing the effectiveness of Q-RM on generalization ability.
- Also suggest testing on more OOD benchmarks (e.g., AIME, LiveCodeBench) to assess generalization.
- Explore smaller QRM backbones (e.g., 7B/13B) paired with policy models of comparable size. Using 70B QRMs limits practical applicability and raises scalability concerns.

**Essential References Not Discussed:**

N/A

**Experimental Designs Or Analyses:**

I suggest:

- Include **cross-domain evaluations** (e.g., coding, long-form QA) to test generality.
- Compare QRM with **RewardBench**-calibrated models to isolate reward modeling improvements.

**Methods And Evaluation Criteria:**

- **Model Sizes**: The choice of 70B QRM backbones across all experiments lacks justification. Smaller QRMs (e.g., 7B) paired with same-scale policies would better demonstrate efficiency.
- **Policy Ablations**: Standardizing to newer models (e.g., Llama-3-8B-Instruct) compared to 1B choice would be better.

**Other Comments Or Suggestions:**

N/A

**Other Strengths And Weaknesses:**

**Strengths**:

- Novel theoretical grounding for token-level credit assignment Q-RM.
- Strong empirical results across multiple tasks.

**Weaknesses**:

- Limited scalability analysis (QRM/policy size mismatch).
- Narrow benchmark scope.

**Questions For Authors:**

1. Why is KL divergence incorporated into advantage calculation (line 238) rather than adding KL loss on $L^{CLIP}$.
2. Can QRM and policy models share weights (e.g., via a 7B backbone) to reduce memory overhead? Like actor and critic sharing most weights.
3. Have you evaluated QRM on **RewardBench** to quantify its reward modeling accuracy vs. baselines?

**Relation To Broader Scientific Literature:**

The proposed token-level Q-RM seamlessly integrates into existing reinforcement learning frameworks, beating previous token-level RMs or sequence-level RMs and PRMs.

**Theoretical Claims:**

The theoretical framework is compelling but would benefit from:

- A step-by-step proof linking Proposition 3.3 to practical advantage equivalence.
- Discussion of approximation boundaries (e.g., when the linear relationship breaks down).

---

> ### Author Rebuttal · Authors · 2025-04-01
>
> Thank you for your thoughtful review and valuable feedback. Below, we address your concerns in detail.
>
> > Q1: Regarding  equivalence claim in Proposition 3.3 and assumptions.
>
> A1: We address it from two key aspects:
>
> - According to **Proposition 3.3**, the optimal Q-function and the optimal logits from Q-RM differ only by a constant offset. The left-hand side of Eq. 18 is exactly the definition of the advantage function, using $Z^*(s_t, a_t)$ to estimate the advantage is equivalent to using $Q^*(s_t, a_t)$.
>
> - According to **Proposition 3.2**, we assume that trajectory entropy $\mathcal{H}^*(\tau)$ is close to zero, which constrains $\mathcal{V}(\tau)$ within a narrow range. This allows Q-RM to be effectively trained using Eq. 17.
>
> We will incorporate these clarifications into revised paper.
>
>
> > Q2: Regarding the effectiveness of Q-RM on generalization ability.
>
> A2: Thank you for your valuable suggestion.
>
> In response, we assess Q-RM's OOD generalization by training ORM, DPO-RM, and Q-RM on the MATH pairwise dataset and using OOD dataset Math10K [1] as RL training instructions. Reward models are based on LLaMA-3-70B-Instruct, while policies are based on LLaMA-3.2-3B-Instruct. As shown in the table below, Q-RM consistently outperforms ORM and DPO-RM, demonstrating its effectiveness in handling OOD scenarios. We will incorporate these OOD results in the updated paper.
>
> | Method | GSM8K Pass@1 | ASDiv Pass@1 | SVAMP Pass@1 |
> |-|-|-|-|
> | REINFORCE+ORM | 66.19 | 83.05 | 83.06 |
> | REINFORCE+DPO-RM | 64.14 | 82.72 | 83.20 |
> | REINFORCE+Q-RM | 69.67 | 84.01 | 84.88 |
>
> [1] LLM-Adapters: An Adapter Family for Parameter-Efficient Fine-Tuning of Large Language Models. EMNLP 2023.
>
> > Q3: Regarding the model size of Q-RM.
>
> A3: We address it from the following perspectives:
> - **Model Size of Q-RM.** It is well established that larger models are more effective for ORM compared to smaller ones. Moreover, this work focuses on training a fine-grained reward model for RL training, which presents a greater challenge than ORM. To ensure broad effectiveness across different policy optimizations, a robust backbone for the reward model is essential. Therefore, we selected the 70B model as the backbone for Q-RM.
>
> - **Additional Experiments.** To further analyze the impact of model size on Q-RM, we use LLaMA-3-8B-instruct as the backbone for both Q-RM and policy, comparing it against ORM and DPO-RM. As shown below, Q-RM remains effective with smaller architectures but excels with the 70B backbone, supporting our choice. Additional results will be added to revised paper.
>
> | Method | GSM8K Pass@1 | MATH Pass@1 |
> |-|-|-|
> | SFT | 63.68 | 30.37 |
> | REINFORCE+ORM (8B) | 71.57 | 34.53 |
> | REINFORCE+DPO-RM (8B) | 72.81 | 33.41 |
> | REINFORCE+Q-RM (8B) | 74.28 | 38.00 |
> | REINFORCE+Q-RM (70B) | 79.98 | 41.23 |
>
> > Q4: Regarding KL loss on $\mathcal{L}^{CLIP}$.
>
> A4: While some works (e.g., [1]) incorporate KL divergence directly into the final training objective $\mathcal{L}^{CLIP}$, it is more common practice [2][3][4] to integrate it into the trajectory rewards. Therefore, in the PPO+Q-RM training framework, we followed this widely adopted approach.
>
> [1] Deepseekmath: Pushing the limits of mathematical reasoning in open language models. ArXiv 2024.
>
> [2] REINFORCE++: A Simple and Efficient Approach for Aligning Large Language Models. ArXiv 2025.
>
> [3] Delve into PPO: Implementation matters for stable RLHF. NeurIPS Workshop 2023.
>
> [4] https://github.com/huggingface/trl
>
>
>
> > Q5: Can QRM and policy models share weights?
>
> A5: Yes, Q-RM can share a backbone with the policy model to save memory, but this limits generalizability and reduces learnable parameters, potentially weakening reward signal capture. Thus, we chose separate backbones for Q-RM and the policy model.
>
>
> > Q6: Have you evaluated QRM on RewardBench to quantify its reward modeling accuracy vs. baselines?
>
> A6: Q-RM focuses on finer-grained rewards for RL training, unlike ORM's sequence-level rewards, so we did not originally test it on RewardBench. However, to address this concern, we evaluated Q-RM against ORM and DPO-RM using RewardBench.
>
> As shown in the table below, Q-RM performs worse than ORM and DPO-RM in terms of pairwise accuracy. This result aligns with Q-RM’s objective, which **prioritizes fine-grained reward modeling and improved policy optimization over maximizing pairwise accuracy**.
>
> Furthermore, high pairwise accuracy in reward models does not necessarily translate to better policy optimization [2]. This is further illustrated in Figure 2(b) of our paper, where Q-RM, despite having the lowest pairwise accuracy, achieves the highest policy optimization performance.
>
> | Method | Chat | Chat Hard | Safety | Reasoning |
> |-|-|-|-|-|
> |ORM|96.63|61.84|78.78|87.21|
> |DPO-RM|94.69|59.39|77.84|84.67|
> |Q-RM|92.91|56.14|71.41|82.21|
>
> We will include the results and discussion in the revised version of the paper.
>
> [1] What Makes a Reward Model a Good Teacher? ArXiv 2025.

---

### Official Review · Reviewer_AzLu · 2025-03-13

**Overall Recommendation:** 2

**Summary:**

When training LLMs with RL, it matters to have fine-grained credit assignment to tokens. The current popular algorithms like GRPO and RLOO all assign the same credit to all tokens in a response. This does not seem ideal at all. There have been recent works in improving this. Q-RM suggests that we compute these fine-grained credits based on the logits of another LLM, a discriminative policy, which is basically trained on preference data in case of RLHF tasks or correct vs incorrect responses in mathematical reasoning tasks with DPO. They show that the logits of a discriminative policy can provide credits to generated tokens and therefore providing better training signal for RL training of LLMs.

**Claims And Evidence:**

The main claim of the paper: Using Q-RM to provide token level credit enhanced RL training of LLMs.
Evidence: Increased pass@1 performance in GSM8K and MATH.

However, I have some concerns regarding if the evidence supports the claim.
1-Based on Nvidia's reported numbers for Llama-3.2-3B-Instruct, it seems the model already achieves pass@1 48.0 accuracy on MATH dataset as reported https://build.nvidia.com/meta/llama-3.2-3b-instruct/modelcard. However, in the paper the highest reported achieved accuracy on MATH is reported as near 35%. This is lower than the initial model. What is the reason for this discrepancy? That is my main criticism towards the results as it suggests something fundamental may be missing in evaluation or loading these models. Am I understanding this correctly?

I have also a concern about the claim. The Q-RM is a 70B model that will undergo training with DPO. I don't have a problem with that. However, I am wondering if the 70B model is being somehow distilled to the small model.

**Essential References Not Discussed:**

There is a paper I remember that discusses the importance of credit assignment when doing RL for LLMs which I think is quite related: VinePPO: Unlocking RL Potential For LLM Reasoning Through Refined Credit Assignment. This paper shows that improving the credit assignment by estimating the advantage via multiple MC rollouts really enhanced RL for LLM training.

**Experimental Designs Or Analyses:**

The only criticism I have is why the authors went for the 70B model as their reward model. This raises questions about distilling the capacities of the 70B model to the smaller model. I understand that other methods that Q-RM is compared to are given the same advantage though. However, I still cannot understand what was wrong with using the same model trained on preference datasets as the Q-RM?

**Methods And Evaluation Criteria:**

I think the benchmarks are reasonable. However, I am not much a fan of averaging GSM8K performance and MATH performance in the tables. GSM8K is significantly simpler than MATH and averaging results on these two is not a very sensible thing to do. However, that is a minor criticism. In general, these two are the most important for reasoning. The field is changing as they are getting saturated but they are fine for now.

**Other Comments Or Suggestions:**

I don't have a comment.

**Other Strengths And Weaknesses:**

I think I listed all the things I have in mind in the above.

**Questions For Authors:**

I have listed my questions already. However, I will list them again for easy reference:
Q1) Is there any reason to go for the 70B model as the RM to train a 7B model? Can't you just use the same base model for both RM and Policy? I know the 7B is the instruct, but the instruct could be also fine-tuned with DPO. Isn't that correct?
Q2) Is there any good reason for the feac that pass@1 reported on the Nvidia is much higher than what is reported in the paper?

**Relation To Broader Scientific Literature:**

There is huge interest in RL for LLMs now and how we can improve them. The literature is adopting methods in RL to improve the current algorithms as what is currently SOTA in RL for LLM is actually quite simple compared to ideas tested in RL setups. Therefore, I think this work is very relevant to the current literature.

**Theoretical Claims:**

I have not checked them for correctness but they make sense to me. It makes sense that under an optimal discriminative policy, the logits actually provide good rewards for the tokens as the optimal discriminative policy can basically tell where the tokens are incorrect. However, the paper argues this is kind of a credit assignment. I am a bit confused by this. In my mind, credit assignment mostly refers to correct estimation of advantages. However, in this sense, the Q-RM is providing rewards which mostly indicates whether the reported results are incorrect or correct and it is less about the advantage of a token. I hope the authors can clear this up for me.

---

> ### Author Rebuttal · Authors · 2025-04-01
>
> Thank you for your thoughtful review and valuable feedback. Below, we address your concerns in detail.
>
> > Q1: Regarding results discrepancy on MATH dataset.
>
> A1: We clarify that the discrepancy in results for Llama-3.2-3B-Instruct on MATH can be attributed to the **differences in evaluation frameworks**. Nvidia's reported results for Llama-3.2-3B-Instruct are based on Meta's official evaluation framework, as detailed in [documentation](https://github.com/meta-llama/llama-models/blob/main/models/llama3_2/eval_details.md). While our paper adopts OpenCompass, which is widely used in recent research [1~3]. Below, we outline the key differences between these two frameworks:
>
> - **Use of LLM-as-Judge**:  Meta's evaluation employs an extra LLM to act as a judge, using an `equivalence template` to determine whether two expressions are equivalent. This introduces an external dependency into the evaluation process. OpenCompass does not rely on an extra LLM for evaluation.
>
> - **Use of Latex Parser**:  Meta's framework utilizes Sympy to parse and verify the correctness of LaTeX equations. OpenCompass does not incorporate such parsing mechanisms, which can lead to lower reported accuracy due to stricter evaluation criteria.
>
> - **Prompting and Sampling Strategy**:  Meta's framework employs zero-shot CoT prompting with top-p sampling, while we adopt OpenCompass to use a zero-shot greedy decoding without additional CoT templates, eliminating the influence of different prompting strategies and randomness on the evaluation.
>
> In summary, we selected OpenCompass for its widespread use in recent studies [1-3], ensuring comparability. In response to the reviewer, we are working to reproduce the results using Meta's framework. Since this requires retraining the policy models, it will take some time, and we will provide an update within 48 hours.
>
> [1] Compression Represents Intelligence Linearly. COLM 2024.
>
> [2] Internlm2 technical report. Arxiv 2024.
>
> [3] Internlm-math: Open math large language models toward verifiable reasoning. Arxiv 2024.
>
>
> > Q2：Regarding Q-RM training with DPO. Is the 70B being distilled into the smaller model? Use same base model for both RM and Policy.
>
> A2: Thank you for your thoughtful comments. We address them from the following perspectives:
> - **Q-RM is Not Trained Using DPO.** We would like to clarify that Q-RM is not trained using DPO. In this work, Q-RM is trained using the proposed discriminative policy, as outlined in Equation 17 of the paper. This approach explicitly decouples reward modeling from language generation. Q-RM and DPO both use preference data but differ in how they model token-level rewards.
>
> - **Model Size of Q-RM.** It is well established that larger models are more effective for ORM compared to smaller ones, and training fine-grained reward models for RL is even more challenging. To ensure broad effectiveness across different policy optimizations, a robust backbone for the reward model is essential. Therefore, we selected the 70B model as the backbone for Q-RM.
>
> - **Relationship with Knowledge Distillation.** Our approach follows standard reward modeling in RL, focusing on fine-grained supervision for policy optimization. Unlike knowledge distillation, we don't directly distill the reward model into the policy.
>
> - **Additional Experiments.** To further analyze the impact of model size on Q-RM, we use LLaMA-3-8B-instruct as the backbone for both Q-RM and policy, comparing it against ORM and DPO-RM. As shown below, Q-RM remains effective with smaller architectures but excels with the 70B backbone, supporting our choice. Additional results will be added to revised paper.
>
> | Method | GSM8K Pass@1 | MATH Pass@1 |
> |---|---|---|
> | SFT | 63.68 | 30.37 |
> | REINFORCE+ORM (8B) | 71.57 | 34.53 |
> | REINFORCE+DPO-RM (8B) | 72.81 | 33.41 |
> | REINFORCE+Q-RM (8B) | 74.28 | 38.00 |
> | REINFORCE+Q-RM (70B) | 79.98 | 41.23 |
>
>
> > Q3: Regarding the averaging results on GSM8K and MATH.
>
> A3: We agree that GSM8K is simpler than MATH, so averaging their results isn't ideal. Our original intent was to report overall results on math domain. To avoid confusion, we will present separate results for GSM8K and MATH in Table 1 without averaging their performance.
>
> > Q4: Regarding the credit assignment in Q-RM.
>
> A4: We acknowledge that credit assignment primarily pertains to the estimation of advantages, and we agree that Q-RM does not directly compute advantages. However, as demonstrated in Proposition 3.3, the rewards generated by Q-RM maintain a linear relationship with the optimal Q-function. This relationship enables the estimation of advantages using Q-RM. We will clarify how Q-RM facilitates credit assignment in the revised paper.
>
>
> > Q5: Regarding the related paper.
>
> A5: Thank you for referencing VinePPO. We recognize its key insight on accurate credit assignment in RL fine-tuning of LLMs and will cite it while discussing its findings in our revised paper.

---

> > ### Comment · Reviewer_AzLu · 2025-04-09
> >
> > Thank you for your detailed answers. I understood that I did not understand the method correctly. Given the fact that I really tried to understand it, I think maybe the method section of the paper is not written clearly. I read the paper again and the equation 17, and I still don't think one can understand from the text that that equation is the objective. I think the paper could have be written much more clearly.
> >
> > I appreciate the effort that you put into your reply and it makes me feel bad not to increase my score. However, I still think the writing of the paper is the main bottleneck:
> > 1-Writing about what the actual method is. It seems very obscured. I remember I really tried to understand it and then my understanding was wrong and I am not that much of a lazy reader.
> > 2-Writing on why going for a bigger model to train the smaller model is OK in experiments. I think it is better to explain it as at its current form it raises concerns.
> >
> > Thank you again.

---

> > > ### Author Response · Authors · 2025-04-09
> > >
> > > Thank you for your thoughtful feedback. We agree that the clarity of the methodology section is crucial. To address your concerns more comprehensively, we provide detailed responses below:
> > >
> > > **Regarding Clarifying the Methodology**
> > >
> > > We plan to make the following revisions in the final version of the paper:
> > > + We will restructure Section 3.3 to explicityly state that Equation 17 represents the final training objective for Q-RM.
> > > + We will add a subsection to outline the overall training objective for Q-RM before diving into implementation details.
> > >
> > > **Regarding Using Larger Model to Train Small Model**
> > >
> > > Our explanation is as follows:
> > > + Larger models are more effective for reward modeling compared to smaller ones. And in this work, we focus on training a fine-grained, token-level reward model for RL training, which presents a greater challenge than outcome-based reward modeling. The increased difficulty arises because token-level reward modeling requires evaluating and assigning rewards at a much finer granularity (at the level of individual tokens), rather than assessing the overall outcome of a sequence. This makes the modeling process more complex and demanding. To ensure robust and broad effectiveness, we select 70B model as backbone for all reward models in the experiment.
> > > + This design also meets the current growing demand for lightweight models deployment. While large models excel in performance, they require high computational costs. By training smaller policies to align with larger reward models, we aim to achieve inference efficiency while approximating the performance ceiling of large models.
> > >
> > > We will incorporate this discussion into the revised version.
> > >
> > > **Regarding Evaluating Results Using Meta's framework**
> > >
> > > Additionally, we provide updated results for Llama-3.2-3B-Instruct on the MATH-500 benchmark using Meta's framework, as shown in the table below. These results exhibit a similar trend to Table 1 in the paper, demonstrating that the improvement achieved by our method is independent of the evaluation framework.
> > >
> > > | Method | MATH-500 Pass@1 |
> > > | --- | --- |
> > > | SFT | 47.01 |
> > > | REINFORCE+DPO-RM | 48.54 |
> > > | REINFORCE+Q-RM | 51.94 |
> > >
> > > We hope these revisions and clarifications address your concerns effectively. Thank you again for your valuable feedback!

---

### Decision · Program_Chairs · 2025-05-01

**Decision:**

Accept (poster)

**Comment:**

The paper introduces a Q function reward model for credit assignment in RL for LLMs. The reviewers appreciate both the theoretical and empirical contributions. Some concerns are raised about the evaluation setting and technical details, most of which have been addressed by the additional experiments and clarifications by the authors. Besides the promised revisions, I encourage the authors to give more hand-holding to readers not familiar with RMs and RL.